# Can Dermoscopy Be a Useful Follow-Up Tool in Patients with Discoid Lupus Treated with Anifrolumab?

**DOI:** 10.3390/diagnostics15050522

**Published:** 2025-02-21

**Authors:** Francesca Ambrogio, Caterina Foti, Silvia Noviello, Gerardo Cazzato, Alexandre Raphael Meduri, Carolina Marasco, Angelo Vacca, Benedetta Tirone

**Affiliations:** 1Section of Dermatology and Venereology, Department of Precision and Regenerative Medicine and Ionian Area (DiMePRe-J), University of Bari “Aldo Moro”, 70124 Bari, Italy; caterina.foti@uniba.it (C.F.); meduri.alexandre@gmail.com (A.R.M.); benedetta.ti96@gmail.com (B.T.); 2Internal Medicine ‘Guido Baccelli’, Department of Precision and Regenerative Medicine and Ionian Area (DiMePRe-J), University of Bari ‘Aldo Moro’, AUOC Policlinico di Bari, 70124 Bari, Italy; silvia.noviello88@gmail.com (S.N.); carolina.marasco@policlinico.ba.it (C.M.); angelo.vacca@uniba.it (A.V.); 3Section of Molecular Pathology, Department of Precision and Regenerative Medicine and Ionian Area (DiMePRe-J), University of Bari “Aldo Moro”, 70121 Bari, Italy; gerardo.cazzato@uniba.it

**Keywords:** anifrolumab, systemic lupus erythematosus, DLE, dermoscopy

## Abstract

This report discusses a female patient with longstanding discoid lupus erythematosus (DLE) and systemic lupus erythematosus (SLE), refractory to multiple immunosuppressive and biologic treatments. Upon presenting with infiltrated, hypertrophic plaques in facial and décolletage regions, she was started on anifrolumab therapy after the histopathological confirmation of DLE. Following three infusions, significant clinical and dermoscopic improvements were observed, including the resolution of plaques and regression of scarring areas. This case highlights anifrolumab’s efficacy in severe lupus skin manifestations, emphasizing its potential to induce dermoscopic and histological remission. Additionally, it suggests that dermoscopy could be a valuable tool for monitoring therapeutic responses in DLE and cutaneous lupus erythematosus, warranting further investigation.

Figure 1. We report the case of a 55-year-old female patient referred to our clinic in June 2023. She had been affected by DLE since 2000 and SLE with joint involvement since 2017, confirmed by multiple biopsies and laboratory tests. Her medical history also included the following: patent foramen ovale, hepatic steatosis, uterine polyps, spondyloarthrosis, and left convex lumbar scoliosis. Over the years, the patient had undergone various immunosuppressive and biologic treatments without benefit, including topical corticosteroids, systemic corticosteroids, hydroxychloroquine, cyclosporine, methotrexate, and belimumab.

At her first visit to our center, the patient presented with infiltrated and hypertrophic brownish (Figure 1A; red arrow) plaques with a peripheral erythematous halo (Figure 1A; red circle) in the eyebrow, zygomatic, nasal, and décolletage regions. Dermoscopy revealed typical findings of DLE [1,2,3,4,5], particularly in the eyebrow region, including the following: speckled brown pigmentation (Figure 1B; red circle), keratotic plugs (Figure 1B; red arrow), vessels (Figure 1B; rectangle), perifollicular halo, background erythema (Figure 1B; triangle), and areas of scarring fibrosis (Figure 1B; star).

In conclusion, the clinical, dermoscopic, and laboratory findings strongly suggested DLE with systemic manifestations. The lack of response to conventional therapies led us to perform a biopsy of a plaque in the zygomatic region, which confirmed our diagnostic hypothesis. The patient was then started on anifrolumab therapy.

Anifrolumab is a human monoclonal antibody that targets the interferon alpha type 1 receptor. In 2021, the drug received approval for use in patients with moderate-to-severe systemic lupus erythematosus (SLE) who are unresponsive to standard therapies, with a recommended dose of 300 mg intravenously (IV) every 4 weeks. Over the past three years, anifrolumab has proven to be an effective, rapid, and safe treatment for SLE [6], and has also contributed to improvements in associated skin conditions such as discoid lupus erythematosus (DLE) and intractable alopecia [7] caused by DLE [8].

Figure 2. At the follow-up visit, after three infusions of the drug, a significant improvement in the clinical skin condition was observed. The plaques had completely resolved, leaving only slight residual hyperpigmented spots, particularly on the face (Figure 2A; arrow). However, the most notable improvement was evident through dermoscopy, as all previously listed signs of DLE had regressed, including the scarring areas, leaving only a residual hyperpigmented and erythematosus area (Figure 2A; circle). Also, at follow-up after more than one year of treatment, there was still no recurrence of the lesion, confirming, even at dermoscopy, that it occurs without any sign of pathological activity. Moreover, the patient did not experience any adverse effects. This clinical case provides valuable insights. First, it contributes to the growing literature on anifrolumab, which has proven to be an extremely effective treatment for SLE, leading to rapid and lasting improvement in patients for whom multiple prior therapies have failed. It also highlights the rapid action of anifrolumab on lupus erythematosus skin lesions, preventing further spread and mitigating the severity of the condition [8,9,10]. According to the most widely accepted hypotheses, a defect in Treg cells is believed to be at the origin, resulting in immune system dysregulation with the excessive production of various types of IFN, particularly IFN-α, along with Th1/Th2 inflammatory cytokines [11]. In this context, anifrolumab has already emerged as a promising new therapeutic option for DLE [9]. The expression of type I interferon is increased in patients with discoid lupus erythematosus compared to healthy controls [12]. In advanced cases of alopecia due to DLE, inflammatory infiltration leads to the irreversible loss of hair follicle stem cells. Since follicular structures do not completely disappear in early or mild cases, early intervention is important before the development of scarring alopecia, although the condition is often resistant to treatment [13]. Documented cases show that anifrolumab had positive effects after the first administration, even in patients refractory to local treatments such as corticosteroids, tacrolimus, injections, hydroxychloroquine, oral prednisone, immunosuppressive drugs, belimumab, and Janus kinase inhibitors [14,15]. It is indeed essential to extend the follow-up observation period to thoroughly assess the long-term effects and potential side effects of the treatment. A more extended observation will provide a clearer understanding of the durability of the treatment’s benefits and any possible adverse reactions that may arise over time. This consideration is crucial for ensuring the overall safety and effectiveness of the treatment in the future. 

Most importantly, this case provides dermoscopic documentation of the improvements, which reflect underlying histological changes. Dermoscopic findings further support the regression of the histopathological substrate of lupus erythematosus skin lesions. To date, the dermoscopic monitoring of lupus lesions during anifrolumab therapy has not been reported, and this area warrants further investigation. In this particular case, the most striking finding was the regression of scar tissue, which had traditionally been considered irreversible. The continued use of dermoscopy to monitor outcomes during anifrolumab therapy may help define specific patterns in DLE and cutaneous lupus erythematosus, providing insights into the drug’s efficacy and potentially allowing for predictions of treatment outcomes.

## Figures and Tables

**Figure 1 diagnostics-15-00522-f001:**
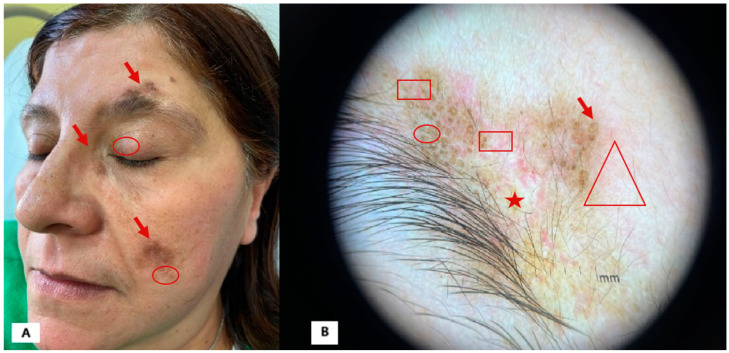
(**A**): Infiltrated brownish and hyperkeratotic plaques on the face; (**B**): dermoscopic picture shows follicular whitish/yellowish keratotic plugs over erythematous background, linear–irregular vessels, and scarring areas.

**Figure 2 diagnostics-15-00522-f002:**
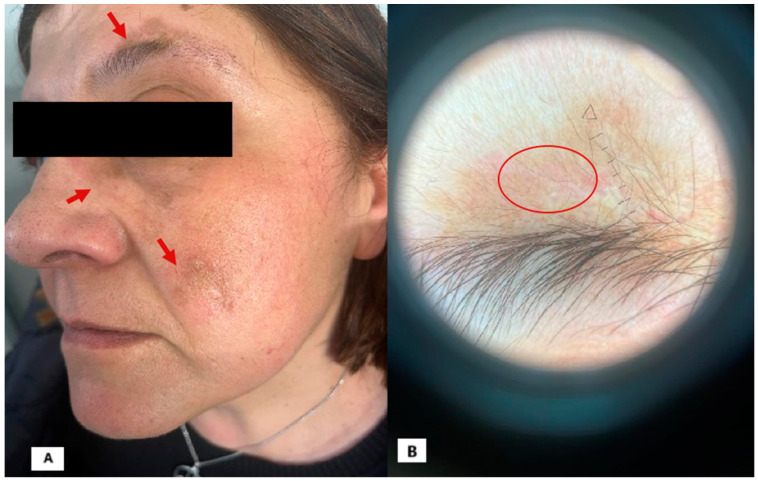
(**A**): Hyperpigmented sequelae of previous facial injuries; (**B**): dermoscopic examination reveals only hyperpigmented areas lacking fibrosis, vessels, and other signs of disease activity.

## Data Availability

The data presented in this study are available on request from the corresponding author due to privacy reasons.

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
