# Peer review of "Can Dermoscopy Be a Useful Follow-Up Tool in Patients with Discoid Lupus Treated with Anifrolumab?"

_diagnostics, 2025, doi:10.3390/diagnostics15050522_

Round 1
Reviewer 1 Report
Comments and Suggestions for Authors.very nice scientific study. As far as we know, anifrolumab has not been used in patients with DLE.It has been successful in proving its effect dermoscopically. (DLE) The response to hydroxychloroquine and topical steroids is good. It should be added to the text whether this response is present or not.This drug may be a hope for patients with generalized DLE.This study may shed light on future studies.
Author Response
Comment 1: .very nice scientific study. As far as we know, anifrolumab has not been used in patients with DLE.It has been successful in proving its effect dermoscopically. (DLE) The response to hydroxychloroquine and topical steroids is good. It should be added to the text whether this response is present or not.This drug may be a hope for patients with generalized DLE.This study may shed light on future studies
Response 1: Thank you for your comment and feedback. You are correct that, to date, anifrolumab has not been widely studied or used in patients with Discoid Lupus Erythematosus (DLE). We appreciate your observation regarding its dermoscopic effects. I apologize for forgetting to mention it in the text, but in our case, neither hydroxychloroquine nor topical corticosteroids led to a good response. This highlights the potential of anifrolumab, which could be considered in cases of resistant forms of DLE. We agree that this study may indeed shed light on the future potential of this drug in treating generalized DLE.
Reviewer 2 Report
Comments and Suggestions for Authors
This study explores the efficacy of Anifrolumab in the treatment of discoid lupus erythematosus and suggests that dermoscopy may serve as a valuable tool for monitoring response to treatment, providing a new perspective in the field of dermatology.
Areas for improvement:
- Currently, only the fact that the patient is female is mentioned. It is recommended to appropriately add key information such as the patient's age and disease duration.
- There is limited exploration of the mechanism. Although significant clinical and dermoscopic improvements have been demonstrated, there is relatively little discussion on how Anifrolumab specifically affects the disease process and the reversal of scar tissue.
- When reporting on the treatment with new biological agents, the evaluation of related adverse reactions or safety is also crucial. It is recommended to make appropriate references in the article.
- Although it is mentioned that no recurrence was observed during the one-year follow-up, more time for follow-up observation is needed regarding the long-term effects and possible side effects.
Comments on the Quality of English Language
The English could be improved to more clearly express the research.
Author Response
- Currently, only the fact that the patient is female is mentioned. It is recommended to appropriately add key information such as the patient's age and disease duration.
Thank you for bringing this to my attention. I have revised the text to include the patient's age and disease duration to ensure greater clarity and completeness. Let me know if you have any further suggestions. - There is limited exploration of the mechanism. Although significant clinical and dermoscopic improvements have been demonstrated, there is relatively little discussion on how Anifrolumab specifically affects the disease process and the reversal of scar tissue.
I have included the information you requested and you were right in highlighting those aspects as they are indeed important elements to consider. - When reporting on the treatment with new biological agents, the evaluation of related adverse reactions or safety is also crucial. It is recommended to make appropriate references in the article.
The suggestion is well taken, and the information has been included to note that, fortunately, no adverse events were experienced by the patient during the treatment. Thank you for highlighting this important point.
- Although it is mentioned that no recurrence was observed during the one-year follow-up, more time for follow-up observation is needed regarding the long-term effects and possible side effects.
Thank you for highlighting this important aspect of the evaluation process. I have added a sentence to emphasize this point and ensure its clarity in the context of the study.